# The Synergistic Activity of Rhamnolipid Combined with Linezolid against Linezolid-Resistant *Enterococcus faecium*

**DOI:** 10.3390/molecules28227630

**Published:** 2023-11-16

**Authors:** Qingru Chang, Huinan Chen, Yifan Li, Hai Li, Zaixing Yang, Jiankai Zeng, Ping Zhang, Junwei Ge, Mingchun Gao

**Affiliations:** 1College of Veterinary Medicine, Northeast Agricultural University, Harbin 150030, China; a06200004@neau.edu.cn (Q.C.); chenhuinan0501@gmail.com (H.C.); S220601043@neau.edu.cn (Y.L.); lihai010308@gmail.com (H.L.); s220602075@neau.edu.cn (Z.Y.); jiankai@neau.edu.cn (J.Z.); zhangpingp@neau.edu.cn (P.Z.); gejunwei@neau.edu.cn (J.G.); 2Heilongjiang Provincial Key Laboratory of Zoonosis, Harbin 150030, China

**Keywords:** linezolid resistance, *Enterococcus faecium*, rhamnolipid, combination therapy, synergistic effect

## Abstract

Enterococci resistance is increasing sharply, which poses a serious threat to public health. Rhamnolipids are a kind of amphiphilic compound used for its bioactivities, while the combination of nontraditional drugs to restore linezolid activity is an attractive strategy to treat infections caused by these pathogens. This study aimed to investigate the activity of linezolid in combination with the rhamnolipids against *Enterococcus faecium*. Here, we determined that the rhamnolipids could enhance the efficacy of linezolid against enterococci infections by a checkerboard MIC assay, a time–kill assay, a combined disk test, an anti-biofilm assay, molecular simulation dynamics, and mouse infection models. We identified that the combination of rhamnolipids and linezolid restored the linezolid sensitivity. Anti-biofilm experiments show that our new scheme can effectively inhibit biofilm generation. The mouse infection model demonstrated that the combination therapy significantly reduced the bacterial load in the feces, colons, and kidneys following subcutaneous administration. This study showed that rhamnolipids could play a synergistic role with linezolid against *Enterococcus*. Our combined agents could be appealing candidates for developing new combinatorial agents to restore antibiotic efficacy in the treatment of linezolid-resistant *Enterococcus* infections.

## 1. Introduction

Enterococci are normal Gram-positive bacteria found in the intestinal tract of humans and animals, which are recognized by their exceptional adaptability to a wide range of harsh environmental conditions [1]. The most representative species are *Enterococcus faecalis* and *Enterococcus faecium* [2]. *E. faecium* has long been used as a probiotic; however, as a conditionally pathogenic bacteria, its multiple drug resistance makes it spread in hospitals worldwide [3], posing a serious threat to public health. It can cause meningitis [4], bacteremia [5], sepsis, and urinary system and soft tissue infections, and these infections are associated with biofilms [6]. *E. faecium*, as a member of the “ESKAPE” pathogens [7], is now regarded as a classic multi-drug-resistant (MDR) pathogen owing to both its incredible ability to generate acquired resistance during antimicrobial chemotherapy as well as its intrinsic resistance to various antibiotics [8].

Effective therapy for infections caused by *Enterococcus* is currently limited [9]. As Gram-positive bacteria, enterococci have a thick cell wall, leading to impaired drug penetration and, thus, inherent resistance to antibiotics. Enterococci have evolved resistance to both aminoglycosides and β-lactams antibiotics as a result of immunosuppression and antibiotic abuse, which has contributed to a rise in the frequency of clinical infections and mortality. Vancomycin-resistant enterococci (VRE) have been detected more frequently since they were first identified and are considered to be second only to *Staphylococcus aureus* as significant nosocomial infection pathogens [10]. In recent years, the use of linezolid (LNZ) as an antibiotic of “last resort” in the clinical treatment for treating infections caused by VRE has increased [11]. However, the high-dose use of LNZ has driven the rapid development of linezolid-resistant enterococci (LRE), and there is a lack of effective antimicrobial tools [12]. Hindered by high costs and time, the drug development pipeline has not kept up with the growing demand for effective novel antibiotics [13]. The steady increase in antibiotic resistance, combined with a decline in the development of novel antibiotics, is threatening to return the world to a pre-antibiotic era, which would be disastrous [14]. To address this issue, the combination of antibiotics with adjuvants or antimicrobials chosen from nature’s reservoir of bioactive substances has been proposed as a new therapy strategy for overcoming bacterial drug resistance [15].

Rhamnolipids (RLS) are surface-active chemicals that are glycolipid biosurfactants mostly produced from Pseudomonas aeruginosa [16]. RLS are composed of hydrophilic and hydrophobic segments linked by covalent bonds, and their unique structure has demonstrated excellent emulsification and decontamination performance. RLS has drawn a lot of attention in recent years for its biological activities against tumors [17], fungi [18], and biofilms [19]. Additionally, studies have revealed that it has an impact on the regulation of the humoral and cellular immune systems [20]. Not only that, RLS do not produce toxicity, including genotoxicity, mutagenicity, subchronic toxicity, and immunotoxicity [21]. However, its potential application for enterococcal infection treatment has not been explored yet.

In the present study, we evaluated the antibacterial and anti-biofilm effects of LNZ combined with RLS against linezolid-resistant enterococci with different strains, including *E. faecium* 83-1B, *E. faecium* 82-2A, and *E. faecium* Y1-8a in vitro. To investigate the mechanism of the synergistic antibacterial effects of LNZ and RLS, we used the molecular docking method to predict the binding ability of RLS to proteins encoding drug resistance genes of *Enterococcus* and detected the effect of both drugs on the expression of drug resistance genes by qPCR assay. Furthermore, we established a mouse *Enterococcus* infection model; explored the antibacterial ability of LNZ and RLS in vivo; detected the weight change; examined feces, colon, and kidney bacterial loads; examined kidney and colon pathological changes; and observed and recorded the clinical symptoms of mice.

## 2. Results

### 2.1. Rhamnolipid Restores the Sensitivity of Linezolid-Resistant Enterococci to Linezolid In Vitro

To evaluate the potential efficacy of RLS, checkerboard dilution assays between RLS and LNZ were performed, and we discovered that RLS potentiated LNZ activities against LRE. As shown in Figure 1A–C, the checkerboard MIC results showed that RLS at the concentrations ranging from 32 μg/mL to 256 μg/mL had a significant synergistic effect with LNZ in the tested LRE, indicating that RLS could restore the sensitivity of all tested strains to LNZ. MIC of RLS was >256 μg/mL for all tested strains, and no influence of RLS (<128 μg/mL) on the growth was observed in the tested bacteria (Figure 1D–F), showing the lack of any intrinsic antimicrobial activity of RLS alone against these LRE isolates. The FICI values were 0.25, 0.5, and 0.25 for the combined action of LNZ and RLS against *E. faecium* 83-1B, *E. faecium* 82-2A, and *E. faecium* Y1-8a, respectively. As a result, the combination of LNZ and RLS has a synergistic effect, with RLS potentially increasing LNZ’s antibacterial activity against most LRE.

To investigate this enhanced effect further, we used the time–killing assays to assess the potentiation effect of RLS with LNZ. As expected, time–kill curves revealed an enormous variance between LNZ + RLS and LNZ groups, with the number of bacteria in the LNZ + RLS group being extremely significantly lower (*p* < 0.01). Furthermore, the combined disk test was used to assess the bactericidal activity of RLS in combination with LNZ, with the inhibition diameter on agar plates being measured. In comparison to 10 μg of LNZ alone, the inhibition zone of 32 μg/mL RLS plus 10 μg LNZ is significantly greater (*p* < 0.01) for the tested isolates (Figure 2).

To evaluate the capacity of RLS and LNZ to prevent the growth of biofilms and destroy tested strains’ premade mature biofilms, the standard crystal violet quantification method was used. As shown in Figure 3A,C, compared with the RLS or LNZ-alone group and the control group, RLS (32–128 μg/mL) combined with LNZ (32 μg/mL) could effectively inhibit biofilm formation in more than half of the *E. faecium* 83-1B and *E. faecium* Y1-8a strains (*p* < 0.01). However, as shown in Figure 3B, the combining of RLS and LNZ seemed to have no discernible removal impact on the developed biofilms of *E. faecium* 82-2A compared to the LNZ group.

### 2.2. Rhamnolipid Is Directly Involved in the Inhibitory Activity of poxtA

To investigate the effect of LNZ and RLS on the expression of resistant genes in *Enterococcus spp*, the expression of mRNA levels of three resistance genes of *E. faecium* 82-2A was examined after 24 h of co-culture with drugs. As shown in Figure 4A, the *poxtA* gene expression of strain 82-2A under the combined effect of LNZ and RLS was significantly lower than that of the LNZ group (*p* < 0.01), whereas the *optrA* gene and *Esp* gene expression were significantly higher than that of the LNZ group (*p* < 0.01).

We performed molecular docking using the Autodock program, and the results showed that rhamnolipid had a docking affinity of less than 0 kJ/mol for the linezolid-resistant gene molecule. The general consensus is that the lower the energy, the more stable the ligand and receptor conformation is, and the greater the possibility of the effect. This demonstrated that rhamnolipid had a high affinity for the poxtA, Esp, and optrA proteins. The analysis of molecular docking results revealed that the binding energy of rhamnolipid to poxtA was −9 kJ/mol, and its docking sites were GLU700, VAL4450, THR410, and LYS410. The binding energy to Esp was −8.9 kJ/mol, with the docking sites ASN438 and LYS86, and rhamnolipid binds with optrA at the active site having a binding energy of −7.8 kJ/mol. The conformation was stable. Figure 4B–D depicts the specific docking site for the docking results.

### 2.3. Rhamnolipid with Linezolid Showed a Synergistic Impact In Vivo Compared to Monotherapy

Based on the remarkable synergistic bactericidal efficacy in vitro of LNZ and RLS, we attempted to confirm whether they have therapeutic potential in vivo. A mouse abdominal cavity infection model infected with *Enterococcus* strains was used to investigate the combinational effectiveness (Figure 5). The clinical results revealed that, except for the mice in group C, which showed no obvious changes, the mice in the other four groups showed chilling, piling up, slow movement, disheveled coat, and loss of appetite beginning at 6 h after infection with enterococci. The first mouse in the infected group died after 7 h, and the second mouse died after 7.5 h. The mice in the LNZ + RLS group resumed their normal diet and activities at 7.5 h, while the mice in the RLS and LNZ groups did so at 8 and 9 h, respectively. To assess general health, we monitored the health states and weight of the mice for 5 days before and after the infection. As shown in Figure 5A, the average body weight of mice treated with LNZ and RLS was significantly higher than that of the infected group (*p* < 0.01) and significantly different compared to the LNZ group (0.01 < *p* < 0.05), and there was no significant difference in body weight between the LNZ and RLS groups.

As in Figure 5, we evaluated the bacterial load in the feces, colon, and kidney of mice. Although monotherapy with LNZ and RLS was effective compared to the control group (Figure 5B–D), enterococci counts were significantly lower in the LNZ + RLS group. The combinational therapy of LNZ and RLS displayed about a 10-fold reduction in bacterial loads in the colon and kidney (*p* < 0.001) and, encouragingly, an almost 100-fold reduction in fecal bacterial load compared to the control group. The histological structures of the combined treatment and control groups were relatively normal, as shown in the pathological sections (Figure 5E), whereas the other groups showed varying degrees of edema and inflammatory cell infiltration in the intestinal epithelial tissue, as well as degeneration of renal tubular epithelial cells and the appearance of bacteria-like granules in the glomerular capillaries. These findings suggested that RLS, in combination with LNZ, significantly protected mice from *Enterococcus* infection, and it could be used as an adjuvant in the treatment of bacterial infections caused by linezolid-resistant pathogens.

## 3. Discussion

Antibiotic resistance in ESKAPE pathogens has become one of the most serious global health threats, contributing to increasing nosocomial infections and deaths [22]. *Enterococcus* can cause a variety of severe life-threatening diseases in humans, as well as a significant risk to farm animals [23,24], wildlife [25], and pets [26]. Vancomycin-resistant enterococci have been a problem for humans for decades, and linezolid, the last line of defense against drug-resistant enterococci, has been compromised. Analysis of the association between antibiotic-resistant bacterial colonization/infection using the WHO Access, Watch, and Reserve (AWaRe) classification revealed that LNZ had the highest correlation with the infection caused by the colonization of critical and high-priority multidrug-resistant organisms (MDROs) [27]. This change in drug resistance may be related to the world’s climate change; it has been demonstrated that drug resistance evolves as a result of intraspecific competition during the facilitation of non-genetic resistance [28]. Traditional antibiotics are no longer effective in preventing and controlling multidrug-resistant enterococci, and new protocols for the control of linezolid-resistant enterococcal infections are urgently needed to address the major threat of enterococcal infections. Many reports have shown that linezolid works synergistically with other antibiotics, thus providing a better fight against infections [29]. The combination of compounds other than antibiotics with antibiotics to form new treatment regimens has garnered considerable scientific attention. In this study, we found that LNZ combined with RLS is an effective therapeutic for linezolid-resistant *E. faecium*, which may be related to the inhibition of optrA gene expression. To the best of our knowledge, this is the first time that RLS and LNZ have been used in combination to address enterococcal infections, thus providing a new therapeutic option for clinical linezolid-resistant enterococcal infections. Although the pharmacological effect of RLS in vivo requires further investigation, RLS as an agent has the potential to be a promising leading compound for clinical infection by LNZ-resistant enterococci in the future.

We noticed that the MIC of LNZ-resistant enterococci was reduced by a factor of four, and the FICI index was between 0.25 and 0.5 (Figure 1A–C). Given that FICI ≤ 0.5 is considered synergistic, we believe that RLS could play a synergistic role with LNZ against *Enterococcus.* The results were similar to the findings for *S. aureus* [30]. This therapy, however, is more broad-spectrum and synergistic against LNZ-resistant enterococci strains than it is against Staphylococcus aureus. In Figure 1D–E, the colony-forming units (CFU) of *E. faecium* in the co-culture of RLS and LNZ decreased significantly, reflecting the significant bactericidal effect of this combination therapy. And in Figure 2, the diameter of the inhibition zone around the LNZ disks with added RLS (32 μg/mL, 64 μg/mL) is larger. A similar phenomenon was observed in Sotirova A’s study [31], where the combination of rhamnolipids and thiosulfonic easters had a synergistic effect in reducing the bactericidal and fungicidal concentrations of MTS and ETS. Curves of *E. faecium* at each RLS concentration did not differ much from each other but were higher than the curve of 83-1B, which was significantly suppressed by RLS at 128 μg/mL (Figure 1G–I). The inhibitory ability of RLS was fairly weak. At whatever dose, RLS had no discernible effect on the other strains. However, RLS significantly enhanced the inhibitory activity of LNZ, and the circles of inhibition produced by the RLS group were all significantly larger than those of the LNZ group, and the inhibitory ability was correlated with the concentration of RLS. The effect of the combination of RLS and LNZ had synergistic anti-*Enterococcus* was favorable and was stronger than LNZ alone. The findings of our study increase the usable life of LNZ, allowing it to be employed in enterococcal infections in the future.

We evaluated the anti-biofilm activity of RLS, LNZ alone, and the combination of RLS and LNZ (Figure 3). For *E. faecium* 83-1B and *E. faecium* Y1-8a, the combined use of LNZ and RLS resulted in a significant reduction in biofilm production. The capability of enterococci to build biofilms complicates the management of enterococcal infections and impedes drug penetration [32,33]. Biofilms are a major cause of chronic infections and device infections [34], putting a significant strain on the healthcare system. In addition to being challenging to treat, biofilm-associated enterococcal infections act as a nidus for bacterial spread and a repository for antibiotic-resistance genes. Enterococci’s ability to build biofilms boosts their treatment resistance and favors their prolonged infection and pollution of the environment and food sector [6]. Several *E. faecium* genes, including *Esp* [35], *atlrA* [36], *spx* [37], and *asrR*, are implicated in biofilm production [6]. We guess that the ability of the combination of LNZ and RLS to inhibit enterococci biofilms may be related to the suppression of the expression of these genes. Nevertheless, we also observed that LNZ and RLS did not synergistically inhibit 82-2A from generating biofilms (Figure 3B). Despite the addition of different concentrations of RLS, the amount of biofilm produced by *E. faecium* 82-2A co-cultured with RLS and LNZ was not significantly different compared to that co-cultured only with LNZ. Diverse *E. faecium* strains show heterogeneity in biofilm properties that may impact this [38]. Biosurfactants such as RLS can mediate the disruption of biofilms [18]. Malandrakis et al. [39] showed that Enterobacter sp-transformed RLS has the ability to inhibit biofilm production in *S. aureus*. The potency of their inhibitory effect on biofilm formation may be related to the species of RLS-producing bacteria [40]. Firdose et al. [41] first suggested that RLS has the ability to inhibit biofilm production in multi-drug-resistant bacteria. E Silva et al. [42] tested the effectiveness of RLS to disrupt or remove S. aureus biofilm and found that it could remove up to 89% of the biofilm from milk. Following RLS treatment, the attached bacteria and matrix were also removed. However, the RLS used in this study did not have the ability to inhibit biofilm production by *Enterococcus faecalis* when used alone; thus, the current study shows that the combination of RLS and LNZ can fight biofilm, providing data to support subsequent clinical use.

We modeled the interaction of RLS with proteins encoded by the *Esp*, *optrA*, and *poxtA* genes. There have been some reports on the antibacterial mechanism of LNZ alone [43,44]. Whereas the synergistic mechanism of RLS and LNZ has never been explored. The molecular docking results showed that RLS has a good ability with Esp, optrA, and poxtA (Figure 4B–D). The binding energies of RLS and the three target proteins are all less than −7 kcal/mol, with RLS binding to the poxtA target protein and the Esp target protein being better than the binding to the optrA target protein. We might safely anticipate that RLS binding to proteins encoded by resistance genes reduces the minimum inhibitory concentration of LNZ against enterococci, resulting in a synergistic anti-enterococcal impact of LNZ and RLS. This provides a research direction for the subsequent synergistic antimicrobial action mechanism of RLS and LNZ. Esp are involved in biofilm formation, causing diseases such as endocarditis and urinary tract infections [45]. The *optrA* gene is primarily responsible for bacterial linezolid resistance, and both *optrA* and *poxtA* are transferable linezolid resistance genes that can lead to enhanced bacterial resistance to oxazolidinone and penicillin [46,47]. We also performed gene-level assays for further investigation. Pcr results in Figure 4A showed that when compared with the LNZ group, the LNZ + RLS group showed significantly lower expression of the *poxtA* gene, whereas the expression of the *optrA* and *Esp* genes was significantly higher, suggesting that the synergistic antimicrobial effect of LNZ and RLS is related to the inhibition of *poxtA* gene expression. To the best of our knowledge, it has not been reported that RLS can have such inhibitory effects on *poxtA* expression. Wang et al. suggested that the accumulation of the *poxtA* gene in LRE isolates may lead to an increase in *poxtA* expression, thus providing the bacteria with an adaptive advantage in the presence of linezolid [48]. Considering that the ABC transporter domain-containing protein encoded by *poxtA* is involved in ATP binding, ATP hydrolysis, and transmembrane translocation, it suggested that this decrease in expression may inhibit these cellular activities, thereby mitigating the emergence of enterococci resistance to linezolid. Down-regulation of related gene expression was also seen in the synergistic antimicrobial and antibiofilm studies of Asadi et al. [49]. This could also have an impact on the cell membrane. Previous research found that RLS interactions with *P. aeruginosa* cells caused changes in outer membrane protein and cell surface morphology [50]. In the study by Sadatti et al. on RLS against MRSA [51], an increase in RLS concentration from 30 to 120 mg/mL increased cell membrane permeability by about 20% and led to a decrease in cell viability of about 70%. Increased RLS concentrations increased cell membrane permeability, which decreased cell survival.

To investigate whether the combined use of RLS and LNZ could also produce favorable results in LRE-infected mice, we conducted a series of in vivo exploratory experiments. We regularly tested the body weight of mice in each group, and Figure 5 A shows that the weight of mice in the LNZ + RLS group was extremely, significantly higher than that of other groups, and they also resumed normal diet and activity more quickly. In addition, the bacterial loads of feces, colons, and kidneys in the LNZ + RLS group were lower than those in the infected group or the monotherapy with the LNZ and RLS group (Figure 5B–D). The results of fecal and colonic loads in mice demonstrated that the combination of LNZ and RLS can promote the egress of pathogenic *E. faecium* from the intestinal tract of mice to reduce its colonization. To some extent, these results are consistent with previous studies on RLS secretion facilitating the expression of antimicrobial protein psoriasin in healthy human skin, which can prevent colonization of pathogens in the earliest phase [52]. Subsequently, we collected pathological sections of the LRE-infected mice and performed HE staining (Figure 5E). The intestinal tissue of the LNZ + RLS group was clear and well-demarcated; the mucosal epithelium was structurally intact; the cell morphology and structure were normal; the intestinal glands in the lamina propria were abundant and regularly arranged; there were a large number of cup cells; and there was no infiltration of inflammatory cells, which was similar to that in the pathological sections of Group C. Moreover, the renal tubular epithelium in the LNZ + RLS group had only slight degeneration. We found that the combination of LNZ and RLS could treat colonic edema and injury, renal tubular epithelial cell degeneration in the kidney, inhibit the proliferative growth of *E. faecium* in mouse organs, and attenuate organ pathologic damage. However, in vitro antimicrobial experiments have shown that RLS does not have an antimicrobial effect. So, is the phenomenon we found linked to the fact that RLS can stimulate the body’s immunity and thus protect the body? Further studies need to be carried out to probe this, and if so, it would be consistent with Zhang et al.’s study [53] that RLS can modulate the immune response in rats. As a commonly used biosurfactant, RLS shows a certain level of safety and is classified as category IV toxicity (non-toxic and non-irritant) by the Environmental Protection Agency (EPA) [54]. In earlier animal safety investigations, RLS was tested for acute irritation and skin sensitization [55]. The results revealed that RLS was not toxic, and hematological tests and biochemical analysis also revealed that RLS had no adverse effects on the experimental animals. The present findings provide a novel option to support the treatment of infections of LRE, exhibiting a significant potential for clinical application. The current in vivo data show that combining RLS and LNZ inhibits the pathogenic effects of *E. faecium* and has a favorable therapeutic efficacy in LRE-infected mice, which is synchronized with the results of the in vitro experiments.

There are some limitations of this study. Despite the observed synergism of the combination of rhamnolipid and linezolid, the mechanism of the synergy remains to be explored in greater depth. The effect of the combination on cell morphology should be studied by scanning electron microscopy (SEM) and transmission electron microscopy (TEM). Drug-resistance gene expression changes have been tracked using transcriptome sequencing in drug-resistance reduction research [56]. Additionally, Western blotting has been used to assess how medicines affect the expression and activity of drug-resistance enzymes [57]. As a next step, we intend to further analyze the synergistic antimicrobial mechanism of RLS and LNZ and validate our results through these techniques. At the same time, we will explore the interaction of other classes of antibiotics with rhamnolipid and the effect on other Gram-positive bacteria.

In summary, our study found for the first time that RLS restores enterococci susceptibility to LNZ. In this study, we show that a combination of RLS and LNZ may be an alternative treatment option against linezolid-resistant *Enterococcus*. Furthermore, the effect of rhamnolipid combined with linezolid against these bacteria, as well as against biofilm formation, was validated in this study. However, further study is needed to further clarify the mechanisms of this synergistic activity. The discovery of rhamnolipid as a promising linezolid adjuvant highlights the possibility of biosurfactants in the treatment of bacterial infections.

## 4. Materials and Methods

### 4.1. Bacterial Strains and Chemicals

The bacterial strains used in this study included *E. faecium* 83-1B, *E. faecium* 82-2A, and *E. faecium* Y1-8a, which were isolated and purified from fur-bearing animals by our laboratory [58]. Linezolid was purchased from Sigma (Sigma, St. Louis, MO, USA). Rhamnolipid was purchased from Xian Ruijie Biological Technology Co., Ltd. (Xi’an, China).

### 4.2. Checkerboard Assays

The MIC assays were used to identify synergies between RLS and LNZ against LRE and were carried out using the broth microdilution method in accordance with the Clinical and Laboratory Standards Institute guidelines (CLSI) [59]. Add 160 μL of bacterial suspension (5 × 10^5^ CFU/mL), 20 μL of LNZ, and 20 μL of RLS to each well of a 96-well plate. The bacteria to be tested in 96-well plates were made to incubate with different concentrations of LNZ (0 μg/mL, 2 μg/mL, 4 μg/mL, 8 μg/mL, 16 μg/mL, 32 μg/mL, 64 μg/mL, 128 μg/mL) and RLS (0 μg/mL, 2 μg/mL, 4 μg/mL, 8 μg/mL, 16 μg/mL, 32 μg/mL, 64 μg/mL, 128 μg/mL) at 37 °C for 24 h. The OD values were recorded at 600 nm. Also, the fractional inhibitory concentration index was calculated (FICI). FICI was determined by using the equation FICI = MIC (linezolid in combination)/MIC (linezolid alone), plus MIC (rhamnolipid in combination)/MIC (rhamnolipid alone). To evaluate the biological importance, the experiment was performed at least three times [60].

### 4.3. Growth Curves

The strains of *E. faecium* 83-1B, *E. faecium* 82-2A, and *E. faecium* Y1-8a were grown in LB broth with culturing at 200 rpm and 37 °C until they reached an optical density at 600 nm (OD600) of 0.3, after which they were averaged into four conical flasks containing varying concentrations of RLS, including 0, 16, 32, and 64 μg/mL. The bacteria were then cultivated at 37 °C with shaking for another hour, and growth was assessed by measuring absorbance at 600 nm every 1 h [61].

### 4.4. Time–Killing Assays

Time–kill curves were used to assess the potential bactericidal impact of RLS coupled with linezolid [62]. Activated LRE were inoculated in LB liquid medium and incubated with RLS (32 μg/mL), LNZ (64 μg/mL), RLS (32 μg/mL) in combination with LNZ (64 μg/mL), or a normal control (PBS) for 0, 1, 3, 5, and 7 h at 37 °C until the bacterial concentration was 1 × 10^5^ CFU/mL [63]. At the time points mentioned above, 200 μL of bacterial solution of each group was taken and coated on the surface of the LB agar medium. After incubation at 37 °C overnight, the number of colonies was counted [57]. The results were calculated and plotted.

### 4.5. Combined Disk Test

The combined disk test (CDT) was performed as described by Guo et al. [57]. The tested bacterial strain suspensions (OD600 = 0.1) were dispersed onto LB agar plates, and 10 μg LNZ disks (Oxoid Ltd., Basingstoke, UK) were placed in the centers. Subsequently, 10 μg of various concentrations of RLS (0 μg/mL, 32 μg/mL, and 64 μg/mL) were added to the disks. The diameters of the inhibition zones around the LNZ disks (with or without RLS) were measured and compared following incubation for 24 h at 37 °C. The experiment was repeated three times.

### 4.6. Biofilm Eradication Test

Biofilm formation inhibition test to investigate the effect of RLS in combination with LNZ on the biofilm formation of LRE performed the same as described with minor modifications [64]. The bacterial liquid was diluted to OD600 = 0.4 with TSB liquid medium at 37 °C and diluted to 1 × 10^6^ CFU/mL. The corresponding concentrations of RLS and LNZ were prepared. Add 160 μL of bacterial solution to a sterile 96-well plate, and then add rhamnolipid and linezolid diluted with TSB medium so that the bacteria are mixed with LNZ (16 μg/mL), LNZ (16 μg/mL) + RLS (16 μg/mL), LNZ (16 μg/mL) + RLS (32 μg/mL), LNZ (16 μg/mL) + RLS (64 μg/mL), LNZ (16 μg/mL) + RLS (128 μg/mL) and are co-cultured. A total of 200 μL sterile TSB medium was used as a negative control; 160 μL bacterial solution and 40 μL TSB medium were used as a positive control (group C) and then incubated in a constant temperature incubator at 37 °C for 24 h. After that, the floating bacterial solution was aspirated and washed 2–3 times with PBS and then dried and fixed. Afterward, 0.1% crystalline violet 200 μL was added to each well for 20 min, washed 2–3 times with distilled water, and air-dried [65]. Then, 200 μL of 33% glacial acetic acid was added to each well, shaken for 30 min, and OD590 nm was measured using an enzyme marker; the experiment was repeated three times, and the results were recorded and plotted on a bar graph.

### 4.7. Molecular Docking

The structures of the target proteins (poxtA, esp, optrA) were retrieved from the RCSB Protein Data Bank (http://www.rcsb.org, accessed on 1 February 2023) and saved in PDB format. The PyMOL (version 4.6.0, https://pymol.org, accessed on 3 February 2023) software is used to separate macromolecules and their ligands, and the AutoDock Tools (version 1.5.6, http://autodock.scripps.edu/, accessed on 5 February 2023) repair the charge by removing water molecules, adding hydrogen atoms, and adding Gastge charges to make them receptor macromolecules. The rhamnolipid molecule was drawn by Chemdraw software (version 19.0.0.26, http://chemdraw.com.cn, accessed on 6 February 2023) and was set as a ligand by using the AutoDock tool for hydrogen addition, assigning central atoms, and testing for twistable bonds. The binding energies and binding sites of compounds and macromolecules to proteins were calculated by the genetic algorithm GA, and finally, the results of docking were visualized using PyMOL.

### 4.8. qPCR

qPCR was carried out as described [66]. The activated *E. faecium* 82-2A was inoculated in LB medium and incubated at 37 °C until the concentration was 5 × 10^5^ CFU/mL. At 37 °C, 1600 μL of bacterial solution was incubated with 400 μL of drug solution for 24 h. The control group drug was LNZ (32 μg/mL), and the experimental group drug was LNZ (32 μg/mL) + RLS (32 μg/mL). RNA was extracted using the Total RNA Extraction Kit at a concentration of 0.2 μg/μL. Then, 10 μL of RNA solution was transferred to an RNAase-free EP tube, and 2 μL of random primers were added, mixed well, and cooled rapidly after 10 min in a water bath at 72 °C. The reverse transcription system in Table 1 was spiked and water-bathed to obtain the reverse transcription product cDNA.

This cDNA served as a template, with *tufA* serving as an internal reference gene. Applied Biosystems^®^ 7500 Real-Time PCR Systems were used for the qPCR experiments. The tests were carried out to detect the expression of the *Esp*, *optrA*, and *poxtA* genes using the reaction system and the reaction conditions described following the manufacturer’s instructions. The relative expression levels were calculated using the 2^−ΔΔCT^ method, and histograms were plotted after all experiments were repeated three times.

### 4.9. Animal Studies

Female BALB/c or C57 mice (SPF class) aged 6 to 8 weeks and weighing 18 to 20 g were purchased from the Liaoning Changsheng Biotechnology Co., Ltd. (Shenyang, China). All procedures involving animals and their care were conducted in strict accordance with the requirements of the Laboratory Animal Ethics Committee of Northeast Agricultural University.

### 4.10. Mouse Systemic Infection Model

The BALB/c mice were challenged intraperitoneally with a lethal dose of *E. faecium* 82-2A (5 × 10^8^ CFU) to cause a systemic infection. The experimental mice were randomly divided into five groups of five mice each after adapting to the environment for seven days, including LNZ-treated group (10 mg/kg), RLS-treated group (24 mg/kg), and LNZ + RLS-treated group (LNZ 10 mg/kg, RLS 24 mg/kg), a PBS control group, and a blank control group. Mouse survival was observed until 96 h post-infection. Mice’s body weight and behavior were measured starting on day 0 [67]. Fecal samples or bacterial counts were collected from mice on day 3 to quantify the intestinal microbiota and assess the bacterial concentration in each mouse [68]. Mice were killed by cervical dislocation 72 h after infection, and the colon and kidney of each mouse were collected for bacterial enumeration in bile aesculin agar medium supplemented with 16 μg/mL LNZ, and the above organs were fixed with 10% formaldehyde to produce pathological sections for analysis and observation.

### 4.11. Statistical Analysis

Data results were expressed as mean ± standard deviation (mean ± SD), and one-way analysis of variance (ANOVA) was performed on the main experimental data using SPSS 19.0 software, with *p* < 0.05 indicating significant differences and *p* < 0.01 indicating highly significant differences. Prism 9 (GraphPad, La Jolla, CA, USA) was used to plot the graphs.

## 5. Conclusions

In summary, our study results report the synergistic effect of rhamnolipid in combination with linezolid against *Enterococcus* strains. This combination treatment enhanced the bactericidal effect of linezolid, whereby rhamnolipid could slow down the development of linezolid resistance in clinical strains of *Enterococcus*. The limitations of our study were that we did not explore the effects of combination therapy on cell morphology and how drugs affect the expression and activity of drug-resistant proteins. As a next step, we intend to further analyze the synergistic antimicrobial mechanism of RLS and LNZ and validate our results through more techniques. Research on rhamnolipid as a promising adjuvant to linezolid indicated the possibility of the treatment of bacterial infections. Our cumulative findings may confirm the potential efficacy of rhamnolipid to treat clinically isolated *Enterococcus*.

## Figures and Tables

**Figure 1 molecules-28-07630-f001:**
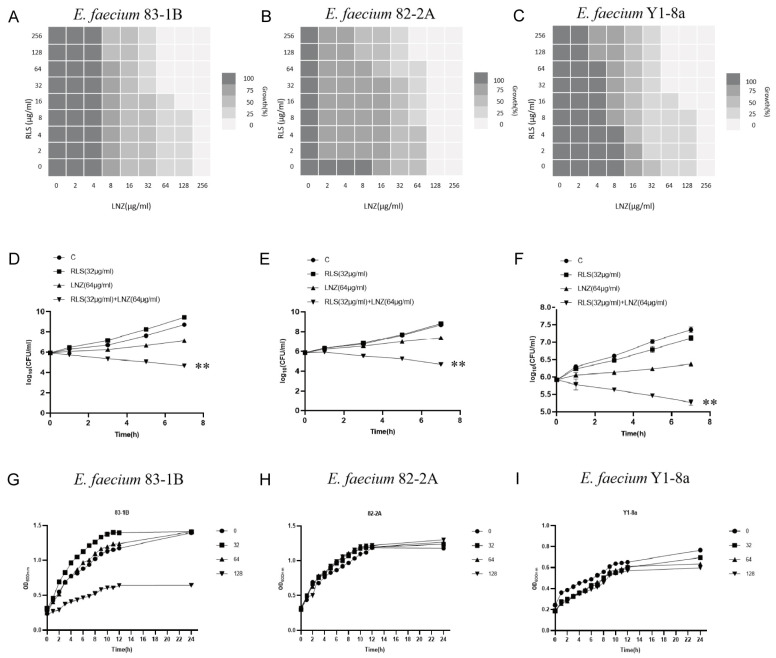
Combined antimicrobial assay results of RLS and LNZ in vitro. (**A**–**C**) Standard broth micro-dilution assay (CLSI) was used to determine the MICs of the combination of RLS and LNZ for *E. faecium* (83-1B, 82-2A, Y1-8a). The FICI values for RLS in combination with LNZ against *E. faecium* 83-1B, 82-2A, and Y1-8a were 0.25, 0.5, and 0.25, respectively. The experiment was performed at least three times. (**D**–**F**) Time–killing curves of RLS and LNZ alone or in combination against three *E. faecium* strains were obtained to evaluate the bactericidal rates. (**G**–**I**) The growth curves of *E. faecium* under different concentrations of RLS (0, 32, 64, 128 µg/mL). Curves of *E. faecium* at each RLS concentration did not differ much from each other but were higher than the curve of 83-1B under RLS at 128 μg/mL. **: *p* < 0.01.

**Figure 2 molecules-28-07630-f002:**
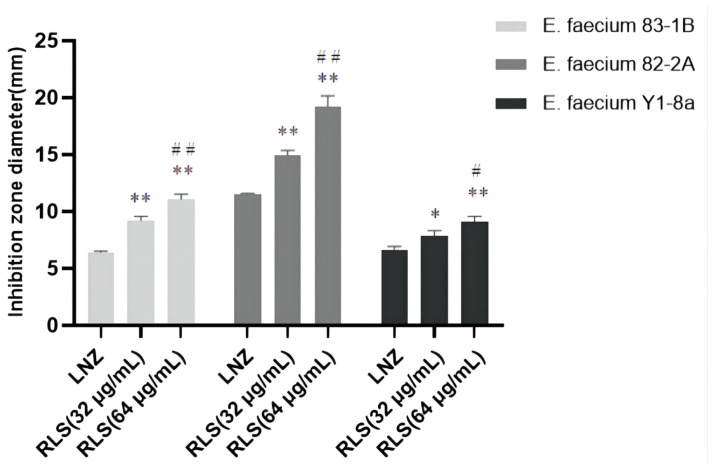
Bacteriostatic circle diameter of the combination of LNZ and RLS on *E. faecium*. Antimicrobial activity of RLS, LNZ, and the combination of RLS and LNZ against *E. faecium* was tested in vitro using the agar disk diffusion method through the measurement of the inhibition zone diameters. The zone of inhibition was significantly increased by the combination of RLS and LNZ. Compared to LNZ group, * denotes a significant difference (0.01 < *p* < 0.05), ** denotes an extremely significant difference (*p* < 0.01); compared to RLS (32 µg/mL) group, # denotes a significant difference (0.01 < *p* < 0.05), ## denotes an extremely significant difference (*p* < 0.01).

**Figure 3 molecules-28-07630-f003:**
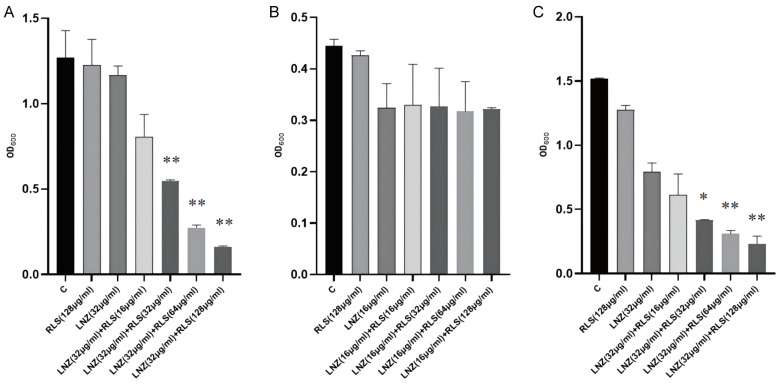
Biofilm production of *E. faecium.* The biofilm production was determined by a crystal violet assay and then visualized by measuring the OD600. RLS had little effect on *E. faecium* biofilm formation, whereas *E. faecium* co-cultured with RLS and LNZ produced less biofilm. The combination of RLS and LNZ significantly inhibited the amount of biofilm formed by *E. faecium*. (**A**) *E. faecium* 83-1B, (**B**) *E. faecium* 82-2A, (**C**) *E. faecium* Y1-8a. *, 0.01 < *p* < 0.05; **, *p* < 0.01.

**Figure 4 molecules-28-07630-f004:**
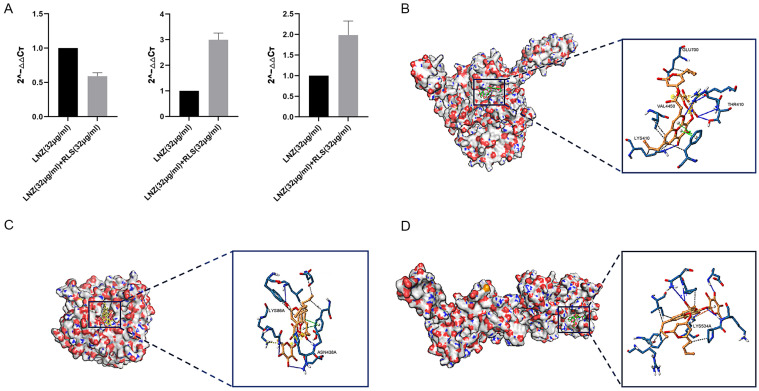
Modeling of RLS binding to linezolid-resistant gene molecule. (**A**) PCR result of the combination of LNZ and RLS on the expression of resistant genes in *E. faecium*. PCR assay was conducted to detect the expression of poxtA, Esp, and optrA. (**B**) Three-dimensional structure determination of poxtA with the RLS complex via molecular modeling method. GLU700, VAL4450, THR410, and LYS410 are docking sites. (**C**) Three-dimensional structure determination of Esp with the RLS complex via molecular modeling method. ASN438 and LYS86 are docking sites. (**D**) Three-dimensional structure determination of optrA with the RLS complex via molecular modeling method. GLU531 and LYS534 are docking sites.

**Figure 5 molecules-28-07630-f005:**
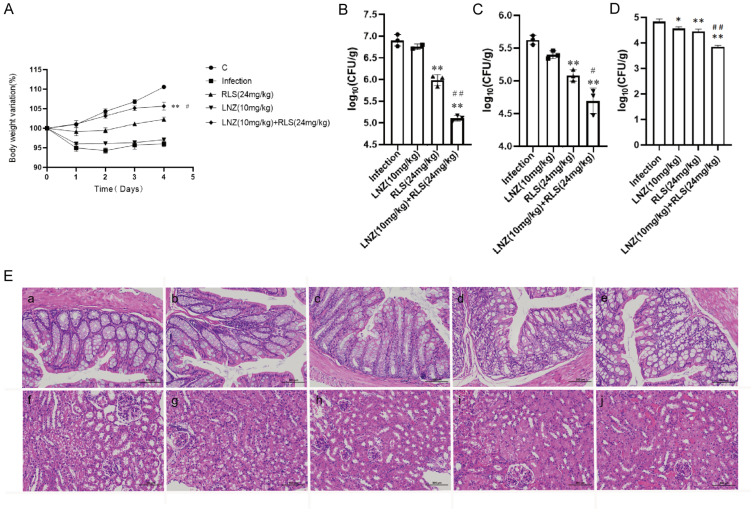
Combined antimicrobial assay results of RLS and LNZ in vivo. Mice were infected with *E. faecium* and then treated with RLS, LNZ, RLS combined with LNZ, or a blank control. (**A**) Average body weight of infected mice. Only mice in the control group and LNZ + RLS group displayed constant weight gain, whereas RLS group and LNZ group mice showed a trend to decrease body weight. (**B**) The bacterial load of enterococci in the feces, (**C**) colon, and (**D**) kidney. Compared to the infection group, * denotes a significant difference (0.01 < *p* < 0.05), ** denotes an extremely significant difference (*p* < 0.01); compared to RLS (24 µg/mL) group, # denotes a significant difference (0.01 < *p* < 0.05), ## denotes an extremely significant difference (*p* < 0.01). ((**E**) **a**–**e**) histological HE staining image of mice colon (200×). ((**E**) **f**–**j**) histological HE staining image of mice kidney. (**a**) Control; (**b**) infection; (**c**) RLS (24 mg/kg); (**d**) LNZ (10 mg/kg); (**e**) LNZ (10 mg/kg) + RLS (24 mg/kg). Scale bar, 100 µm.

**Table 1 molecules-28-07630-t001:** Primers used in the study of gene expression in *E. faecium*.

Gene	Primer	Primer Sequence (5′-3′)	References
*16SrRNA*	27F	AGAGTTTGATCMTGGCTCAG	[67]
	1492R	GGTTACCTTGTTACGACTT	
*gyrB*	UP-1	GAAGTCATCATGACCGTTCTGCAYGCNGGNGGNAARTTYGA	[68]
	UP-2r	AGCAGGGTACGGATGTGCGAGCCRTCNACRTCNGCRTCNGTCAT	
*Esp*	Esp-F	CCACGAGTTAGCGGGAACAG	[22]
	Esp-R	TTGGAGCCCCATCTTTTTCA	
*tufA*	tufA-F	TACACGCCACTACGCTCAC	[22]
	tufA-R	AGCTCCGTCCATTTGAGCAG	
*optrA*	optrA-F	CACTGATTTGAGCAAGCTGTTGGTC	
	optrA-R	TATGGATGGTGTGGCAGCATTGTC	
*poxtA*	poxtA-F	TATGGATGGTGTGGCAGCATTGTC	
	poxtA-R	GGTCGGTATTGTCGGCGTGAAC	

## Data Availability

Data are contained within the article.

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
