# Peer review of "The Synergistic Activity of Rhamnolipid Combined with Linezolid against Linezolid-Resistant Enterococcus faecium"

_molecules, 2023, doi:10.3390/molecules28227630_

Round 1

Reviewer 1 Report

Comments and Suggestions for Authors

It is a well written research paper. 

I only have one query that How to define the dosage of LNZ treated group (10mg/kg), RLS treated group (24 mg/kg) in animal mode.

In addition, there are two suggestions. 1.  Please give more specific finding in these captions of Figure. 2. Because of the title of this manuscript is "Synergistic activity of rhamnolipid combined with linezolid against linezolid-resistant Enterococcus faecium", the limitation of "we did not explore the interaction of other classes of antibiotics with rhamno lipid and the effect on other Gram-positive bacteria"   might be not necessary.

Reviewer 2 Report

Comments and Suggestions for Authors

Dear athours,

I have only few concerns.

The presentation of the figures is good, but in all of your figures any interested readers will read a caption which is a concluding remark. This is contrary to the fact that any caption of a Figure must simply presents the profile(s) in a Figure. Therefore, adjust the captions in all the figures in the manuscript to that of presenting the name of a Figure. Caption should not be used for concluding remarks. Hence, your discussion must perfectly discuss the profiling of each dynamics in each figure. Such confusion in your manuscript has left your Discussion part starting with a motivational remarks. Discussion means to discuss the figures profiles. Then Concluding remarks must be reserved for the concluding remarks in the conclusion of the manuscript. I hope this point is clear. Moreover, your conclusion lacks future direction, please add the research future direction.

Secondly, the significance of the content of the manuscript does not highlight its influence on the effects of climate change, contrary to the current major concern on the topic, worldwide. This a concern to any interested reader of this manuscript. Please in your manuscript, indicate whether or not the resistance addressed by by manuscript has some relationship to the effect of climate change or not.

There has been extensive work done in the direction of the drug resistance which has established the links between the drug resistance and the climate change. Thus, I recommend your manuscript to cite: On the hindering evolution of drug resistance due to intraspecific competition arising during the facilitation survival for non-genetic resistance with fractal fractional, in the manuscript.

Thirdly, any interested reader would want to see the work:

Synergistic activity of rhamnolipid combined with linezolid against linezolid-resistant Enterococcus faecium

Q Chang, H Chen, Y Li, H Li, Z Yang, J Zeng, P Zhang, M Gao, J Ge

bioRxiv, 2023biorxiv.org

cited in the manuscript. If this work cannot be cited in the manuscript, please motivate why? In the same vein, any interested reader would also want to know how is the work

Synergistic activity of rhamnolipid combined with linezolid against linezolid-resistant Enterococcus faecium

Q Chang, H Chen, Y Li, H Li, Z Yang, J Zeng, P Zhang, M Gao, J Ge

bioRxiv, 2023biorxiv.org

different from this manuscript?

Unless the above recommendations are attended to, otherwise, I do not recommend this manuscript for publication in this journal.
